# *CsPHYB*–*CsPIF3*/*4* Regulates Hypocotyl Elongation by Coordinating the Auxin and Gibberellin Biosynthetic Pathways in Cucumber (*Cucumis sativus* L.)

**DOI:** 10.3390/plants14030371

**Published:** 2025-01-26

**Authors:** Liqin Chen, Zongqing Qiu, Jing Dong, Runhua Bu, Yu Zhou, Huilin Wang, Liangliang Hu

**Affiliations:** College of Horticulture, Xinjiang Agricultural University, Urumqi 830052, China; clq20010201@163.com (L.C.); zongqing_qiu@163.com (Z.Q.); dongjing122024@163.com (J.D.); brh2011702533@163.com (R.B.); zy18194938971@163.com (Y.Z.); wanghuilin@126.com (H.W.)

**Keywords:** cucumber, *CsPHYB*, *CsPIF3/4*, auxin, gibberellin, hypocotyl elongation

## Abstract

Hypocotyl length is closely related to quality in seedlings and is an important component of plant height vital for plant-type breeding in cucumber. However, the underlying molecular mechanisms of hypocotyl elongation are poorly understood. In this study, the endogenous hormone content of indole acetic acid (IAA) and gibberellin (GA_3_) showed an increase in the long hypocotyl *Csphyb* (phytochrome B) mutant AM274M compared with its wild-type AM274W. An RNA-sequencing analysis identified 1130 differentially expressed genes (DEGs), of which 476 and 654 were up- and downregulated in the mutant AM274M, respectively. A KEGG enrichment analysis exhibited that these DEGs were mainly enriched in the plant hormone signal transduction pathway. The expression levels of the pivotal genes *CsGA20ox-2,* in the gibberellin biosynthesis pathway, and *CsYUCCA8,* in the auxin biosynthesis pathway, were notably elevated in the hypocotyl of the mutant AM274M, in contrast to the wild-type AM274W. Additionally, GUS staining and a dual-luciferase reporter assay corroborated that the phytochrome-interacting factors *CsPIF3/4* can bind to the E(G)-box motifs present in the promoters of the *CsGA20ox-2* and *CsYUCCA8* genes, thereby modulating their expression and subsequently influencing hypocotyl elongation. Consequently, this research offers profound insights into the regulation of hypocotyl elongation by auxin and gibberellin in response to light signals and establishes a crucial theoretical groundwork for cultivating robust cucumber seedlings in agricultural practice.

## 1. Introduction

In higher plants, following seed germination, hypocotyl elongation represents the earliest light-responsive event in photomorphogenesis [1]. *PHYB* functions as a positive regulator in the photomorphogenesis process and can modulate hypocotyl elongation by mediating the transmission of light signals to downstream elements [2,3,4,5]. Within the nucleus, phytochrome B interacts with phytochrome-interacting factors (*PIFs*), which act as suppressors of photomorphogenesis. This interaction triggers the rapid phosphorylation of PIFs, subsequently leading to their degradation via the 26S proteasome pathway, thereby impeding hypocotyl elongation [6,7,8,9,10]. Additionally, *PHYB* can also interact with the E3 ubiquitin ligase gene *COP1*, which exerts a negative regulatory influence on light signals in a red-light-dependent manner [4]. This interaction augments the accumulation of *HY5*, a positive regulator of light signals, and ultimately promotes the inhibition of hypocotyl elongation [10]. In addition to *PIFs* and *COP1*, recently, a series of transcription regulatory factors associated with photomorphogenesis have been identified. These include *CPK6*, *CPK12*, *SWC6*, *ELF4*, *PRR1/TOC1*, and *BBX28*, which are involved in regulating hypocotyl elongation as downstream constituents of the *phyB* protein [11,12,13,14].

Auxin was the first identified plant hormone, and it can regulate multiple aspects in the process of plant growth and development, including the elongation of the plant hypocotyl [15]. In Arabidopsis, the overexpression of *YUCCA* genes in the auxin biosynthesis pathway leads to an increase in auxin content, which in turn results in the elongation of the *Arabidopsis* hypocotyl [16]. Under high-temperature conditions, *CsYUCCA8* and *CsYUCCA9* are upregulated, which promotes hypocotyl elongation. Under low-temperature conditions, *CsYUCCA10b* increases and *CsYUCCA4* is inhibited, thus influencing the growth of the hypocotyl in cucumber [17]. In a low R:FR light condition, the auxin content of hypocotyls was increased, thus producing a long hypocotyl in pumpkin [18,19]. These results indicate that auxin is affected by other external environmental factors while regulating hypocotyl elongation.

Gibberellin (GA) is involved in plant growth and development including seed germination, fruit development, cell elongation, etc. [20,21,22]. In tomato, loss of function in the GA receptors *GID1a*, *GID1b1*, and *GID1b2* results in strong inhibition of stem elongation and leaf expansion, whereas the effect on the elongation of the primary root is relatively minor [23]. Additionally, in soybean, the interaction of *GmGBP1* with *GmGAMYB* increases the level of gibberellin (GA) to promote height and hypocotyl elongation by activating the auxin-response gene *GmSAUR* [24].

Although phytochrome B, auxin, and gibberellin signals all play important roles in the elongation of the plant hypocotyl, the relevant studies have mainly reported on the model plant Arabidopsis. Our study offers novel insights into their roles in cucumber, an economically important crop species. In this study, we determined that the phytochrome B-mediated light signal can regulate the elongation of the cucumber hypocotyl by modulating the auxin and gibberellin signaling pathways, eventually laying an important theoretical foundation for the cultivation of strong seedlings in cucumber.

## 2. Results

### 2.1. Phenotypic Characterization of Mutant AM274M and Double-Mutant AM274Mlh2

AM274M is a spontaneous mutant with a longer hypocotyl (length = 153 mm) than that in WT AM274W (length = 41 mm) due to a 5.5 kb LTR retrotransposon insertion within the *CsPHYB* gene, encoding a phytochrome B protein (Figure 1) [25]. In the growth chamber, the length of the hypocotyl in the double-mutant AM274M*lh2* (length = 57 mm) is intermediate between that of AM274M and the *lh2* mutant (a GA-deficient mutant, length = 25 mm) (Figure 1C,D). The results indicated that *CsPHYB* is partially epistatic with *lh2* in the process of regulating the elongation of the AM274M hypocotyl. Therefore, it is speculated that in the process of *CsPHYB* regulating the elongation of the AM274M hypocotyl, in addition to GA playing a role, there may also be other factors involved, such as auxin, which was identified as another hormone signaling factor (see below), as well as temperature [25].

### 2.2. GA and IAA Content in Mutant AM274M

Previous studies have demonstrated that plant hormones play a very important role in the process of hypocotyl elongation [26,27,28,29,30]. In order to explore the role of plant hormones in the process of *CsPHYB*-mediated cucumber hypocotyl elongation, the hormone contents in the hypocotyl of the mutant AM274M were determined. As shown in Figure 2, the GA3 and IAA levels were significantly higher in the mutant compared to the WT (Figure 2B,D), while there were no significant differences in ZT, ABA, or JA between the mutant and the WT (Figure 2A,C,E). This indicates that GA_3_ and IAA may be involved in the hypocotyl elongation of AM274M regulated by *CsPHYB*.

### 2.3. Exogenous GA and IAA Promote Hypocotyl Elongation

There are significant differences in GA_3_ and IAA between the hypocotyls of the mutant AM274M and the WT. It is speculated that these two hormones play important roles in the process of *CsPHYB*-mediated hypocotyl elongation in AM274M. For this purpose, 100 μmol GA_3_ and 300 μmol IAA were exogenously added to the MS medium, respectively, to observe their effects on the hypocotyl elongation of AM274M. As shown in Figure 3, compared with the control, under the action of exogenous GA_3_ and IAA, the hypocotyls of AM274M and the WT showed obvious elongation, and the lengths of the hypocotyls increased significantly (Figure 3). These results indicate that GA_3_ and IAA have a promoting effect on the hypocotyl elongation of the mutant AM274M.

### 2.4. Comparative Transcriptome Profiling Analysis

In order to explain the changes in hormone contents in the mutant AM274M and the potential regulatory network of *CsPHYB* in regulating the hypocotyl elongation of AM274M, transcriptome sequencing was carried out on the hypocotyls of the mutant and the wild-type on the seventh day. Then, the transcriptome results were analyzed. A total of 1130 differentially expressed genes (DEGs) were identified. Among them, 476 were upregulated and 654 were downregulated in the mutant when contrasted with the WT (Figure 4A,B). Based on the fact that cell elongation is the direct cause of the formation of the long hypocotyl in AM274M, genes related to cell elongation were sought when screening for differential genes. It was found that *CsEXT3*, *CsEXPA8*, *CsXTR6*, *CsXTH22*, and *CsDWF4* were all upregulated in AM274M (Appendix A). In addition, many transcription factors such as *PIF3/4*, *COP1*, *MYB106*, *bHLH87*, and *TCP15* were also found to be upregulated in AM274M in the list of differential genes (Appendix A). This indicates that these transcription factors may also be involved in regulating the elongation of the AM274M hypocotyl. GO enrichment analysis was performed on the differential genes. In the biological process category, the results mainly included protein–chromophore linkage, response to light stimulus, and circadian rhythm; in the cellular component aspect, they mainly included photosynthetic membranes, photosystem I, and chloroplast thylakoid membranes; and in the molecular function category, they involved tetrapyrrole encoding, chlorophyll encoding, and heme encoding (Figure 4C). The GO enrichment analysis demonstrated the reasons for the occurrence of *CsPHYB* functional defects; that is, abnormalities occurred in relevant processes in the phytochrome synthesis pathway, such as the synthesis of the tetrapyrrole chromophore, an important component of the phytochrome protein, and the linkage between the apoprotein and the chromophore.

KEGG pathway enrichment analysis revealed that the differential genes were mainly enriched in pathways such as plant hormone signal transduction, circadian rhythm, and photosynthesis antenna proteins (Figure 4C). In the hormone signal transduction pathway, the key genes of the GA biosynthesis pathway, gibberellin oxidase *CsGA20ox-2*, *CsGA3ox*, and kaurenoic acid oxidase *CsKAO*, were upregulated in the mutant AM274M, while the GA inactivation gene, gibberellin oxidase *CsGA2ox*, was downregulated. In addition, the expression level of the GA receptor *CsGID1* was upregulated in AM274M, while the repressor DELLA protein *CsGAI1* in the GA signal transduction pathway was downregulated (Appendix A). This indicates that the higher GA content in AM274M is due to an upregulation of gene expression in the GA biosynthesis pathway. Meanwhile, the downregulation of the GA signal transduction pathway repressor gene *CsGAI1* further strengthens the GA signal transduction and thus promotes the elongation of the AM274M hypocotyl. Additionally, the key gene of the IAA biosynthesis pathway, flavin monooxygenase *CsYUCCA8*, and signal transduction pathway genes, such as the auxin-response factor *CsARF19* and auxin-responsive protein *CsIAA16,* were also upregulated (Appendix A). This shows that the upregulation of genes related to the IAA biosynthesis pathway is the reason for the increase in auxin content in AM274M, and the signal transduction enables IAA to further play an important role in the hypocotyl elongation of AM274M.

### 2.5. Expression Analysis of CsGA20ox-2 and CsYUCCA8 Genes

To verify the reliability of the transcriptome sequencing results and further confirm that the IAA and GA signaling pathways are truly involved in the hypocotyl regulatory pathway mediated by *CsPHYB–CsPIF3/4*, we selected *CsGA20ox-2* and *CsYUCCA8* for qPCR in the hypocotyls of the cucumber mutant AM274M on the seventh day. *CsGA20ox-2* is a key gene in the gibberellin biosynthetic pathway, and its expression level is known to significantly influence gibberellin content, which in turn affects hypocotyl elongation. Similarly, *CsYUCCA8* is a crucial gene in the auxin biosynthetic pathway, and its upregulation has been shown to cause increased auxin content and elongated hypocotyls in previous studies [31]. Meanwhile, we also detected the expression levels of the hypocotyl growth-promoting factors *CsPIF3/4* in the mutant AM274M. The results suggested that the expression levels of *CsGA20ox-2*, *CsYUCCA8*, and *CsPIF3/4* in the mutant AM274M were significantly higher than those in the WT, while *CsPHYB* showed the opposite expression trend (Figure 5). In combination with the transcriptome results, it can be implied that *CsGA20ox-2* and *CsYUCCA8* are indeed involved in the process of hypocotyl elongation in the mutant mediated by *CsPHYB–CsPIF3/4*.

### 2.6. CsPIF3/4 Can Bind to the E(G)-Box of the CsGA20ox-2 and CsYUCCA8 Promoters

In order to analyze the potential regulatory mechanism causing the changes in GA and IAA contents in the mutant AM274M, we predicted the binding sites of the *CsPIF3/4* transcription factor within the promoters (2000 bp upstream of the ATG start codon) of the genes *CsGA20ox-2* and *CsYUCCA8*, respectively. The results revealed the presence of four potential E-box elements, namely, P4/CAATTG (−1489~−1494), P3/CAACTG (−1407~−1412), P2/CAAATG (−1072~−1077), and P1/CATATG (−295~−300), within the promoter region of the *CsGA20ox-2* gene. Meanwhile, analysis of the *CsYUCCA8* gene promoter sequence identified a G-box element, P5/CACGTG (−841~−846) (Figure 6A,B).

GUS staining analysis and dual-luciferase reporter assays were performed to verify whether *CsPIF3/4* has the function of regulating *CsGA20ox-2* and *CsYUCCA8* expression. Firstly, the GUS staining results showed that when the effector *35S-CsPIF3* was co-injected into tobacco leaves with the reporter factors *CsGA20ox-2-P1* and *CsYUCCA8-P5*, respectively, the GUS staining turned blue, while no color change was observed in the control. When the effector *35S-CsPIF4* was combined with the reporter factor *CsYUCCA8-P5*, GUS staining also turned the tobacco leaf discs blue (Figure 6C,D). Additionally, the dual-luciferase activity assay results indicated that compared to the control, the LUC/REN activity ratio increased when *CsPIF3* was combined with *CsGA20ox-2-P1* and *CsYUCCA8-P5* and also increased when *CsPIF4* was combined with *CsYUCCA8-P5*. However, when *CsPIF3* was co-transformed with *CsGA20ox-2-P2*, *CsGA20ox-2-P3*, or *CsGA20ox-2-P4* into tobacco, not only did the GUS staining fail to turn blue, but there was also no significant difference in the LUC/REN activity ratio compared to the control (Figure 6E–J). Therefore, in summary, these results suggest that *CsPIF3* can promote the expression of *CsGA20ox-2* by binding to the E-box sequence at P1 and that *CsPIF3/4* can promote the expression of *CsYUCCA8* by binding to the G-box sequence at P5.

## 3. Discussion

In our previous study, the mutant AM274M was reported, with a long-hypocotyl phenotype due to a 5.5 kb LTR retrotransposon insertion within the phytochrome B gene [25]. To elucidate the underlying molecular mechanism by which *PHYB* regulates hypocotyl elongation and coordinates the hormone signaling pathway in the mutant AM274M, we have presented a compelling series of evidence in this study. Firstly, hormone content measurements in the hypocotyl revealed that both GA_3_ and IAA levels were significantly higher in the mutant AM274M compared to the WT, while no significant differences were observed for other hormones between AM274M and the WT (Figure 2). This indicates that there are differences in the auxin and gibberellin metabolism pathways between the mutant AM274M and WT. Subsequently, transcriptome sequencing of the AM274M hypocotyl identified a significant upregulation of genes related to GA and IAA biosynthesis and response (Appendix A). These experimental results provide compelling evidence that the long-hypocotyl phenotype in the cucumber AM274M is a consequence of *CsPHYB*-mediated biosynthesis and signal transduction of GA_3_ and IAA.

In Arabidopsis, *PHYB* plays a negative regulatory role in hypocotyl elongation in seedlings, as evidenced by a series of *Arabidopsis phyB* mutants (*phyB-1/2/3/4/5/6/7/8/9/10*) that all exhibit longer hypocotyl phenotypes compared to the wild-type [32,33,34,35]. Furthermore, *PHYB* involvement in regulating hypocotyl elongation is mostly indirect, primarily functioning within the nucleus by interacting with regulatory factors associated with hypocotyl elongation [11,36,37,38,39,40]. Among the numerous interacting factors of *PHYB*, the phytochrome-interacting factors (*PIFs*), as a class of *bHLH* (basic helix–loop–helix) family transcription factors, play an antagonistic role with *PHYB* in regulating hypocotyl elongation. The *Arabidopsis* T-DNA insertion mutants *pif3* and *pif4* both exhibit shortened hypocotyls, while overexpression lines display elongated hypocotyl phenotypes, indicating that *CsPIF3/4* may have a positive regulatory role in cucumber hypocotyl elongation. Many studies regarding auxin- and gibberellin-regulated hypocotyl elongation are mainly concentrated on the model plant Arabidopsis. In Arabidopsis, the gibberellin biosynthesis-deficient mutant *gal* and the GA response-deficient mutant *gai* both exhibit short hypocotyls. However, after exogenous GA_3_ treatment, the length of the hypocotyls was restored [41]. Regarding flavin monooxygenase-like proteins (*YUCs*), as the rate-limiting enzymes in the auxin synthesis pathway, the *yucca* mutant exhibits a shorter hypocotyl. When *YUCCA* is overexpressed, it leads to an increase in auxin content and also exhibits hypocotyl elongation in Arabidopsis [16]. In this study, when exogenous GA_3_ and IAA were added to the MS medium, the mutant AM274M exhibited a longer hypocotyl than both the control and WT (Figure 3). Furthermore, the AM274M*lh2* double-mutant phenotype provided additional evidence that *CsPHYB* may partially act upstream of the GA synthesis pathway, thereby regulating hypocotyl elongation in the mutant AM274M (Figure 1).

qRT-PCR analysis verified that the expression of *CsGA20ox-2*, a key gene in the gibberellin biosynthetic pathway, and *CsYUCCA8*, a key gene in auxin biosynthetic pathway, were both significantly upregulated in the mutant AM274M compared with the WT in the present study (Figure 5). Previous studies have highlighted the significance of phytochrome-interacting factors (*PIFs*) as crucial transcriptional regulators in integrating light and hormone signals, particularly in controlling hypocotyl elongation [42]. In this study, we employed dual-luciferase reporter assays and GUS staining analysis of promoter activity to demonstrate that *CsPIF3/4* can bind to the promoters of the downstream gene *CsGA20ox-2*, while *CsPIF4* also binds to the promoter of *CsYUCCA8*, thereby enhancing their expression and elevating the endogenous levels of GA and IAA in the mutant AM274M (Figure 6). Furthermore, our study experimentally confirms that *CsPHYB*, GA, and IAA can synergistically regulate hypocotyl elongation in the mutant AM274M, with *CsPIF3/4* serving as the bridge that facilitates this interaction. In the case of hypocotyl elongation, although both cucumber and Arabidopsis involve the interaction of *PHYB* and *PIF3/4*, the specific regulatory details vary. In the present study, we have identified the key roles of *CsGA20ox-2* and *CsYUCCA8* in the biosynthesis pathways of gibberellin and auxin, respectively, and their direct regulation by *CsPIF3/4*. Such a precise molecular mechanism has not been reported in the model plant Arabidopsis.

Therefore, based on the above, and the fact that WT *CsPHYB* but not the mutant *Csphyb* can interact with *CsPIF3/4* as identified by Y2H and BIFC assays in our previous study, we propose the following model to illustrate the potential molecular mechanism by which *CsPHYB* regulates the hypocotyl elongation of the wild-type AM274W and the mutant AM274M by modulating GA and IAA biosynthesis (Figure 7). In the wild-type AM274W with normal hypocotyls, *CsPHYB* can interact with *CsPIF3/4*, resulting in the downregulation of *CsPIF3/4* expression (Figure 5). The binding of *CsPIF4* to the promoter of *CsYUCCA8* and that of *CsPIF3* to the promoter of *CsYUCCA8* and *CsGA20ox-2* are weakened, leading to a decrease in the synthesis content of GA and IAA and ultimately inhibiting hypocotyl elongation. In the mutant AM274M with a long hypocotyl, *Csphyb* cannot interact with *CsPIF3/4*, which leads to the enhanced binding of *CsPIF4* to the promoter of *CsYUCCA8* and that of *CsPIF3* to the promoter of *CsYUCCA8* and *CsGA20ox-2*, thereby promoting the biosynthesis of GA and IAA, respectively, and finally resulting in the hypocotyl elongation of AM274M plants.

Eventually, understanding the molecular mechanism of *CsPHYB*–*CsPIF3/4* in regulating hypocotyl elongation can provide valuable insights for cucumber breeding programs. Breeders can potentially target the genes involved in this pathway to manipulate hypocotyl length and plant height, which are important agronomic traits. In addition, knowledge of this regulatory mechanism can also guide the application of exogenous hormones in seedling management. Based on our findings, the precise application of GA and IAA at specific stages of seedling development could be used to fine tune hypocotyl elongation and improve seedling quality. Moreover, understanding the crosstalk between light and hormone signaling pathways could enable the development of more efficient and sustainable cultivation strategies by integrating light management and hormonal treatments to maximize crop yield and quality. Therefore, this research not only deepens our understanding of the fundamental biological processes underlying hypocotyl elongation in cucumber but also holds great promise for practical applications in improving cucumber breeding and seedling management practices. 

## 4. Materials and Methods

### 4.1. Plant Materials

AM274M is a spontaneous mutant with a long hypocotyl, identified from the cultivated cucumber inbred line AM274W [25]. In the previous study, using a map-based cloning strategy, we identified that a 5.5 kb LTR retrotransposon insertion within the *CsPHYB* gene was responsible for the long-hypocotyl mutation phenotype in AM274M [25]. The GA biosynthesis-deficient mutant *lh2* is a *CsGA20ox-2* (*CsGy6G025610*) loss-of-function mutant but with a normal hypocotyl length, and it was provided by Prof Yiqun Weng’s team at the University of Wisconsin-Madison, USA [31]. The mutant AM274M and wild-type (WT) AM274W were used to perform phenotype analysis, hormone content determination, and transcriptomic sequencing. The *lh2* mutant was used to hybridize with AM274M to obtain the double-mutant AM274M*lh2* to investigate the role of GA signaling in cucumber hypocotyl elongation. All materials were grown in a sunlight greenhouse under natural sunlight at Xinjiang Agricultural University (Urumqi, China).

### 4.2. Extraction and Quantification of Endogenous Hormones

The hypocotyls of the mutant AM274M and WT were taken on the 7th day after germination. For each sample, the hypocotyls of five seedlings were pooled to form one biological replicate, and a total of three biological replicates were prepared for both the mutant and WT.

After the sampling was completed, the hypocotyl was immediately placed into liquid nitrogen for full grinding, 1g of the sample was taken into a 10 mL centrifuge tube, 5 mL of the extraction solvent (Isopropanol: H_2_O: HCl = 2:1:2) was added for shock mixing, and the sample was shaken at 4 °C for 16 h. After centrifugation at 8000 rpm at 4 °C for 10 min, the supernatant was taken into a new centrifuge tube, and two times the volume of dichloromethane was added into the centrifuge tube and shocked at 4 °C for 1h. After the shock was completed, the sample was centrifuged at 12,000× *g* at 4 °C for 15 min.

The lower organic phase was transferred to a distillation flask and distilled at 38 °C. After distillation, the sample was re-dissolved in 1 mL of methanol solution and filtered with a 0.22 μm organic filter membrane, followed by Liquid Mass Spectrometry (QTRAP5500, AB Sciex, Framingham, MA, USA), and was computerized testing; the measurement parameters were set according to the method of Zhang et al. 2024 [43].

### 4.3. Exogenous Hormone Treatment

The seeds of the mutant AM274M and its WT were sterilized and sown on an MS medium containing 3% sucrose (Phytotech, Lenexa, KS, USA). For the treatment groups, 100 μmol GA_3_ and 300 μmol IAA (Solarbio, Beijing, China) were added to the MS medium, respectively. Apart from hormone addition, the other specific operations referred to the method of Zhang et al. [43]. Each treatment was set up with three biological replicates, each consisting of ten seedlings. The control group received the MS medium without any hormone addition. All of the plants were grown under 16 h of light/8 h of darkness at a constant temperature of 25 °C. Hypocotyl data were collected 15 days after germination.

### 4.4. RNA-Seq Analysis

RNA was isolated from the hypocotyls of ten 7-day-old seedlings of both AM274M and the WT using Trizol (Invitrogen, Waltham, MA, USA). Subsequently, we utilized the TruSeq™ RNA sample prep kit from Illumina to construct cDNA libraries, which were then sequenced on the Illumina HiSeq-2500 machine by Majorbio Bio-pharm Technology Co., Ltd., located in Shanghai, China. After filtering out low-quality reads and poly-N sequences, the remaining clean reads were aligned to the reference genome version 9930 v3.0 using TopHat 2.0.13. To quantify transcript abundance, FPKM values were computed by Stringtie, accessible at https://ccb.jhu.edu/software/stringtie/ (accessed on 11 August 2024). Differentially expressed genes (DEGs) were determined based on a significance threshold of an FDR (false discovery rate) of less than 0.05 and an absolute log2 fold change of at least 1. Additionally, GO (Gene Ontology) and KEGG (Kyoto Encyclopedia of Genes and Genomes) pathway enrichment analyses were conducted on the identified DEGs utilizing the Majorbio Cloud Platform, available at https://cloud.majorbio.com/ (accessed on 11 August 2024).

### 4.5. qRT-PCR Analysis

To confirm the regulation of *CsGA20ox-2* and *CsYUCCA8* by *CsPHYB–CsPIF3/4*, the expression levels of *CsGA20ox-2*, *CsYUCCA8*, *CsPHYB*, and *CsPIF3/4* were examined in the hypocotyls of 7-day-old seedlings of the mutant AM274M and WT. Each sample included three biological replicates and three technical replicates to ensure accuracy. For each biological replicate, the hypocotyls of five seedlings were used. The relative expression levels of the target genes were calculated using the 2^−ΔΔCt^ method, which was introduced by Livak and Schmittgen in 2001 [44]. For this calculation, *UBI-ep* (ubiquitin extension protein) was used as the internal reference gene, following the approach described by Wan et al. in 2010 [45]. Detailed information about all the primers used in the qRT-PCR experiments is provided in Appendix A.

### 4.6. Promoter Cis-Acting Element Prediction of CsGA20ox-2 and CsYUCCA8

*CsPHYB* can interact with phytochrome-interacting factors 3 and 4 (*CsPIF3/4*), as confirmed in our previous study using yeast two-hybrid (Y2H) and bimolecular fluorescence complementation (BIFC) assays, respectively. Additionally, given the transcriptome data in this study, there are significant differences between AM274M and WT in the expression of *CsGA20ox-2*, a key gene in the GA biosynthesis pathway, and *CsYUCCA8*, a key gene in the IAA biosynthesis pathway. Therefore, to explore whether IAA and GA signaling are involved in the regulation of hypocotyl elongation in the mutant AM274M, as moderated by *CsPHYB*-mediated light signals, the cis-acting elements E-box (CANNTG) and G-box (CACGTG), which bind to PIFs, are predicted to be present in the promoters of the genes *CsGA20ox-2* and *CsYUCCA8*. For prediction, promoter sequences of approximately 2000 base pairs upstream of the transcription start sites of the two genes were selected for analysis, respectively.

### 4.7. GUS Staining Analysis and Dual-Luciferase Reporter Assay

For GUS staining analysis, each E-box sequence in *CsGA20ox-2* and *CsYUCCA8* was serially repeated three times to form a new sequence, which was then cloned into the vector pCAMBIA121. The recombinant vectors *3×ProCsGA20ox-2-mini35S-Gus* and *3×ProCsYUCCA8-mini35S-Gus* served as reporter factors, and the empty mini35S-GUS was used as a negative control. *CsPIF3/4* was cloned into the vector pCAMBIA3301-EGFP according to our previous study [46] and was used as the effecter factor. All plasmid constructs, after being confirmed as correct through sequencing, were used for the transformation into tobacco (Nicotiana benthamiana) leaves following the method described by Li [47].

For the dual-luciferase reporter assay, approximately 400 bp fragments containing the E-box element regions were cloned into the pGreenII 0800-LUC vector separately, constructing the recombinant vectors pGreenII 0800-*ProCsGA20ox2-LUC* and pGreenII 0800-*ProCsYUCCA8-LUC* as reporter factors. Meanwhile, CsPIF3 and CsPIF4 were cloned into the linearized pGreen II 62-SK vector to construct pGreenII *62SK-CsPIF3/4* as an effector factor vector. Subsequently, three combinations (0800-LUC with 62SK, *ProYUCCA8-LUC* with *62SK-CsPIF3/4*, and *ProGA20ox-2-LUC* with *62SK-CsPIF3/4*) were co-transformed into the pSOUP Agrobacterium, respectively. The mixed bacterial suspension was injected into tobacco leaves, with 10 tobacco leaves injected per combination. After injection, the tobacco plants were placed in a growth chamber for culturing, and samples were collected after 2 days. Using a puncher, 5–6 leaf disks with a diameter of 4–5 mm were collected from the injected areas of the tobacco leaves. Immediately after, the leaf disks were placed into liquid nitrogen for thorough grinding. To the thoroughly ground samples, 100 μL of lysis buffer was added, followed by vortexing to mix evenly and incubation on ice for 5 min. Subsequently, the samples were centrifuged at 12,000 rpm for 1 min, and 20 μL of the supernatant was taken for fluorescence detection.

### 4.8. Statistical Analyses

The obtained data were sorted using Excel 2023. One-way ANOVA was performed with SPSS 20 to compare the hormone contents and qRT-PCR gene expression data between the mutant AM274M and wild-type AM274W, and the significance of differences was detected with the Duncan method. A significant difference was considered at the * *p* < 0.05 and ** *p* < 0.01 levels, and graphs were produced with GraphPad Prism5 software.

## Figures and Tables

**Figure 1 plants-14-00371-f001:**
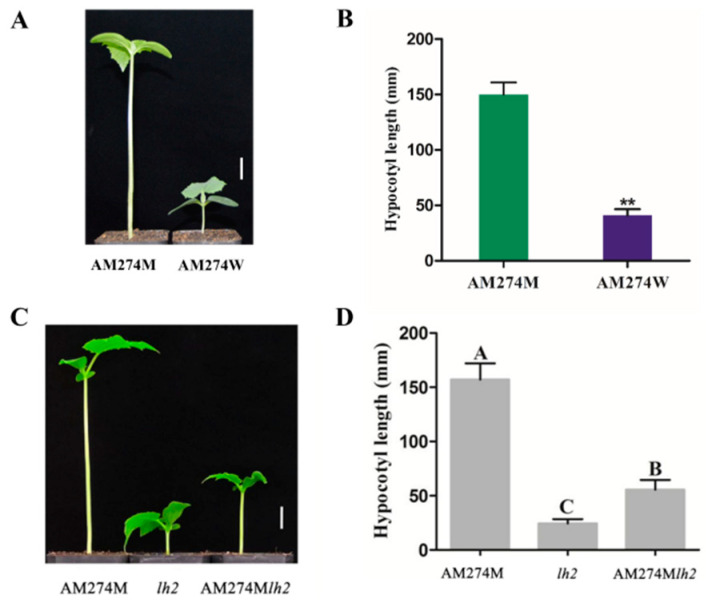
Phenotypic analyses of the long-hypocotyl mutant AM274M. (**A**) Seedling phenotypes of the long-hypocotyl mutant AM274M and its wild-type AM274W. (**C**) Phenotypic characteristics of AM274M, *lh2*, and AM274M*lh2*. (**B**,**D**) The data of the mutant AM274M, wild-type AM274W, mutant *lh2*, and the double-mutant AM274M*lh2* hypocotyl length were collected 15 days after germination, as shown in (**A**,**C**). Scale bar = 2 cm. Data are means ± SD. “**” and the capital letters A–C all indicate statistically significant differences in *t*-tests at *p* < 0.01 between the mutant and WT.

**Figure 2 plants-14-00371-f002:**
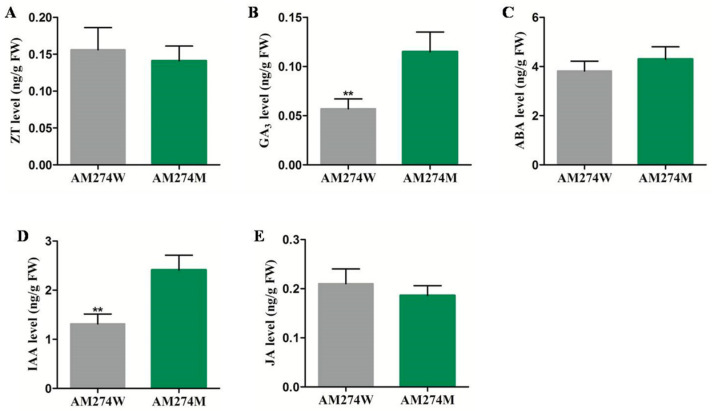
Determination of hormone content in the hypocotyl of the mutant AM274M and wild-type AM274W. (**A**–**E**) ZT, GA_3_, ABA, IAA, and JA content in the hypocotyl of the mutant AM274M and wild-type AM274W. Significance analysis was conducted with two-tailed Student’s *t*-tests (** *p* < 0.01) between the mutant and WT.

**Figure 3 plants-14-00371-f003:**
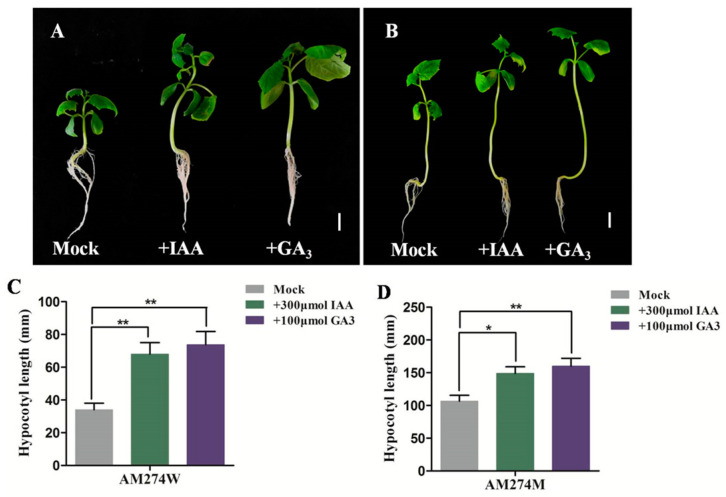
Effects of exogenous GA_3_ and IAA on hypocotyl elongation in the wild-type AM274W and mutant AM274M. (**A**–**D**) Phenotype and length of hypocotyl of the wild-type (**A**,**C**) and AM274M (**B**,**D**) after exogenous GA_3_ and IAA treatment. Scale bar = 2 cm; “*” and “**” denote statistically significant differences in *t*-tests at *p* < 0.05 and *p* < 0.01 between the mutant and WT.

**Figure 4 plants-14-00371-f004:**
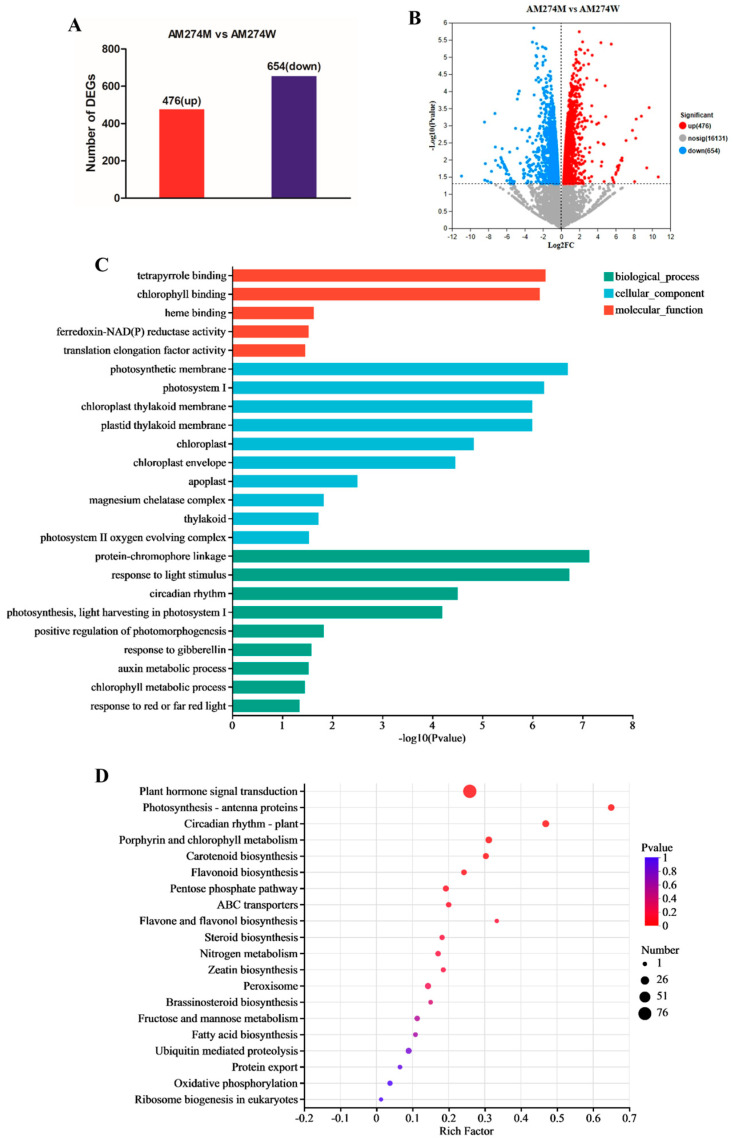
The RNA-seq showed the impact of the AM274M mutation on gene expression. (**A**) The differential number of DEGs between the AM274M mutant and the AM274W wild-type. (**B**) DEG volcano map, where the red dots represent the upregulated genes in AM274M, the blue dots represent the downregulated genes, and the gray dots represent the non-differentially expressed genes. (**C**) GO enrichment analysis of differentially expressed genes between AM274M and AM274W. (**D**) KEGG enrichment analysis of differentially expressed genes between AM274M and AM274W.

**Figure 5 plants-14-00371-f005:**
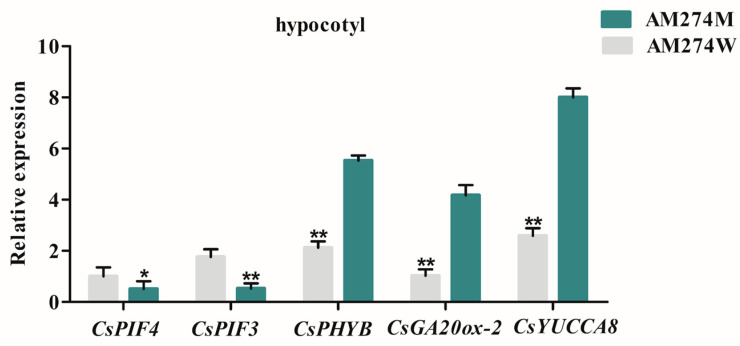
The effects of wild-type AM274W and mutant AM274M on different gene expression levels on the 7th day after germination. Significance level with a two-tailed Student’s *t*-test (* *p* < 0.05 and ** *p* < 0.01).

**Figure 6 plants-14-00371-f006:**
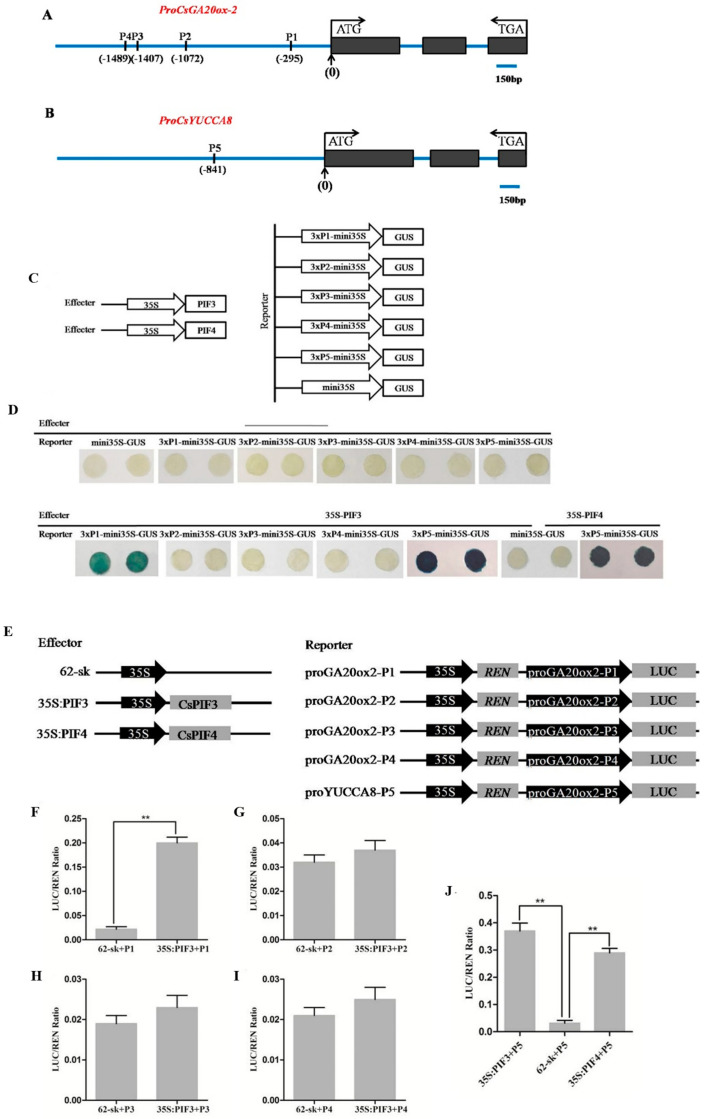
GUS staining analysis and dual-luciferase reporter assay. (**A**,**B**) Prediction of cis-acting elements of the *CsGA20ox-2* and *CsYUCCA8* promoters as a schematic diagram; P1, P2, P3, and P4 represent four different E-box elements, and P5 indicates the G-box element. The black boxes represent gene exons, and the blue lines represent introns and promoters, respectively. (**C**) Construction of effector vectors (*35S:CsPIF3/4*) and reporter vectors (*mini35S-GUS*, *3×GA20ox2-P1/P2/P3/P4-mini35S-GUS*, *3×YUCCA8-P5-mini35S-GUS*). (**D**) GUS staining of tobacco leaves injected with a single reporter and GUS staining of tobacco leaves co-infected with the effector *35S:CsPIF3* and the reporter *3×GA20ox2-P1/P2/P3/P4/P5-mini35S-GUS*, *35S:CsPIF4*, and *3×YUCCA8-P5-mini35S-GUS*. (**E**) Construction of the effector vector *35:CsPIF3/4* and the reporter vector *ProGA20ox2-P1/P2/P3/P4-LUC* and *proYUCCA8-P5-LUC* for dual-luciferase activity assay. (**F**–**J**) *62-SK+P1/P2/P3/P4/P5* means that the empty vector effector was co-injected with the reporter, respectively; *35S:PIF3+P1/P2/P3/P4* indicates that the effector *35S:PIF3* was co-injected with the reporter *ProGA200x2-P1/P2/P3/P4-LUC* and *proYUCCA8-LUC-P5*, respectively; and *35S:PIF4+P5* indicates the co-injection of the effector *35S:PIF4* and *proYUCCA8-LUC-P5*. “**” denotes statistically significant differences in *t*-tests at *p* < 0.01 between the mutant and WT.

**Figure 7 plants-14-00371-f007:**
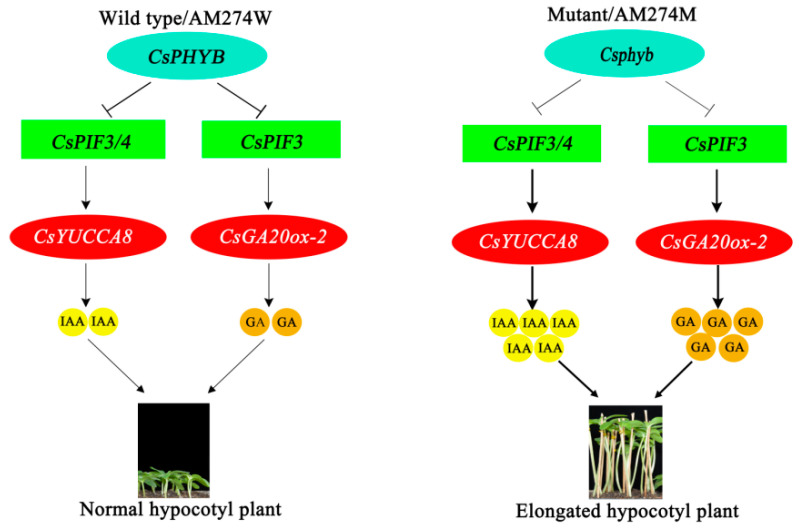
A working model to illustrate the underlying predicted mechanisms of *CsPHYB–CsPIF3/4-*regulated hypocotyl elongation. Line thickness is employed as a quantitative indicator to visually represent the intensity of the promoting effect. Thicker lines indicate a stronger intensity of the promoting effect, whereas thinner lines signify a weaker intensity.

## Data Availability

All of the data that have been generated or analyzed and which underpin the findings of this study are comprehensively incorporated within the main text of the article as well as its Appendix A.

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
