# Peer review of "CsPHYB–CsPIF3/4 Regulates Hypocotyl Elongation by Coordinating the Auxin and Gibberellin Biosynthetic Pathways in Cucumber (Cucumis sativus L.)"

_plants, 2025, doi:10.3390/plants14030371_

Round 1
Reviewer 1 Report
Comments and Suggestions for Authors
The manuscript “CsPHYB-CsPIF3/4 regulates the hypocotyl elongation via coordinates auxin and gibberellin biosynthetic pathways in cucumber (Cucumis sativus L.)” reports a nice overview into PIF and hormone signalling in the cucumber hypocotyl. While most of the reports are on Arabidopsis, this study on cucumber will provide a new insight.
Here are some comments that I wish the authors had addressed to improve the manuscript:
1. Missing references: There are several missing references in the manuscript. Please revise the issue.
2. AM274M has high auxin content. Still, the addition of exgenous auxin causes elongation. Does that mean hypocotyl elongation based on auxin is dose-dependent in cucumber? How does the author explain the saturation level of auxin for hypocotyl growth?
3. Add rationale for choosing these genes for qPCR. Add a piece of brief information in the 2.5 section of the results.
Comments on the Quality of English LanguageEnglish language improvement is required in the introduction, results and discussion parts. In addition, there are several typos that need to be addressed.
Reviewer 2 Report
Comments and Suggestions for Authors
This manuscript investigates the molecular mechanism of hypocotyl elongation in cucumber, focusing on the role of CsPHYB and its downstream targets CsPIF3 and CsPIF4. The study integrates transcriptomic analyses, phenotypic assays, and promoter-binding experiments to demonstrate how CsPHYB-CsPIF3/4 regulates gibberellin (GA) and auxin biosynthesis pathways. The research highlights key findings, such as increased expression of CsGA20ox-2 and CsYUCCA8, and proposes a working model for the regulation of hypocotyl elongation via light-mediated hormonal crosstalk.
This study addresses an important question in plant development and photomorphogenesis, contributing to the understanding of light and hormone signaling in cucumbers. However, significant issues with clarity, novelty articulation, and integration of findings reduce its impact.
- The study focuses on an agriculturally important trait—hypocotyl elongation—and its regulation by light and hormones, which has implications for seedling quality and crop management.
- The combination of genetic, biochemical, and transcriptomic analyses provides a strong foundation for the proposed mechanisms.
- The identification of CsPIF3/4 as regulators of CsGA20ox-2 and CsYUCCA8 expression adds new dimensions to our understanding of light-hormone interaction.
The manuscript does not clearly articulate how this work advances previous studies on PHYB and PIFs in other species. Please clearly state the novelty in the introduction and discussion, emphasizing unique findings specific to cucumber, such as the role of CsPIF3/4 in integrating light and hormone signaling. Compare the results with studies in model plants like Arabidopsis to contextualize the significance.
While the study identifies key genes (e.g., CsGA20ox-2 and CsYUCCA8) regulated by CsPIF3/4, it does not explore additional layers of regulation, such as protein interactions or transcriptional feedback loops. Please investigate the potential interaction of CsPIF3/4 with other transcription factors or cofactors involved in light and hormone signaling. Include chromatin immunoprecipitation (ChIP) to validate promoter binding in vivo.
The results are presented as isolated observations, making it difficult to follow the logical flow of findings. Please integrate the results into a cohesive narrative that connects phenotypic, biochemical, and transcriptomic findings. Use summary diagrams to illustrate the regulatory model and pathways.
The manuscript focuses excessively on fold changes and statistical comparisons without interpreting the broader implications of the findings. Please reduce repetition of numerical data in the text and focus on discussing how these changes impact the physiological and developmental processes of hypocotyl elongation.
The manuscript contains numerous grammatical errors, awkward phrasing, and inconsistent terminology. Example: “The contents of GA3 and IAA in the mutant were significantly higher than those in the WT.” should be: “GA3 and IAA levels were significantly higher in the mutant compared to the WT.” Please conduct thorough language editing for grammatical accuracy and improved readability. Ensure consistent use of scientific terms and proper italicization of gene and species names.
Figures legends are not sufficiently detailed for standalone interpretation. Please ensure all axes, labels, and legends are clear and self-explanatory. Move less critical data to supplementary materials. For example, “he “**” and capital letters A and B indicated statistically significant differences in t tests at P < 0.01.”---the test between what and what?
The discussion is narrowly focused on the findings without considering their implications for cucumber breeding or seedling management. Please expand the discussion to highlight the potential applications of this research in improving seedling robustness and crop yield under different light and hormonal conditions.
Information about biological replicates, statistical methods, and experimental controls is insufficient. Please provide detailed descriptions of the statistical analyses, including the number of replicates, specific tests used, and thresholds for significance.
In second paragraph in Introduction: “In Arabidopsis, Over-expression of YUCCA genes in the auxin biosynthesis pathway leads to an increase in auxin content, which in turn results in the elongation of the Arabidopsis hypocotyl [Error! Reference source not found.].”
Comments on the Quality of English LanguageThe manuscript contains numerous grammatical errors, awkward phrasing, and inconsistent terminology. Example: “The contents of GA3 and IAA in the mutant were significantly higher than those in the WT.” should be: “GA3 and IAA levels were significantly higher in the mutant compared to the WT.” Please conduct thorough language editing for grammatical accuracy and improved readability. Ensure consistent use of scientific terms and proper italicization of gene and species names.
Round 2
Reviewer 2 Report
Comments and Suggestions for Authors
Thanks for the efforts to improve the manuscript.
Author Response
Comments 1: Thanks for the efforts to improve the manuscript.
Response 1: Thank you very much for your kind words and valuable efforts in reviewing our manuscript. Your feedback is greatly appreciated and has been instrumental in enhancing the quality of our work. We are delighted to know that our efforts to improve the manuscript have been recognized. If there are any further suggestions or changes you would like us to consider, please do not hesitate to share them with us. We look forward to continuing to refine our work with your guidance.